# Soft Conductive Hydrogel-Based Electronic Skin for Robot Finger Grasping Manipulation

**DOI:** 10.3390/polym14193930

**Published:** 2022-09-20

**Authors:** Xiao Cheng, Fan Zhang, Wentao Dong

**Affiliations:** 1Rail Transportation Technology Innovation Center, East China Jiaotong University, Nanchang 330013, China; 2Department of Mechanical and Electrical Engineering, Guangdong Polytechnic Normal University, Guangzhou 510665, China; 3School of Electrical and Automation Engineering, East China Jiaotong University, Nanchang 330013, China

**Keywords:** robotic electronic skin (RES), conductive hydrogel, robot finger, grasping manipulation, negative force feedback

## Abstract

Electronic skin with human-like sensory capabilities has been widely applied to artificial intelligence, biomedical engineering, and the prosthetic hand for expanding the sensing ability of robots. Robotic electronic skin (RES) based on conductive hydrogel is developed to collect strain and pressure data for improving the grasping capability of the robot finger. RES is fabricated and assembled by the soft functional materials through a sol–gel process for guaranteeing the overall softness. The strain sensor based on piezoresistive hydrogel (gauge factor ~9.98) is integrated onto the back surface of the robot finger to collect the bending angle of the robot finger. The capacitive pressure sensor based on a hydrogel electrode (sensitivity: 0.105 kPa^−1^ below 3.61 kPa, and 0.0327 kPa^−1^ in the range from 4.12 to 15 kPa.) is adhered onto the fingertip to collect the pressure data when touching the objects. A robot-finger-compatible RES with strain and pressure sensing function is designed for finger gesture detection and grasping manipulation. The negative force feedback control framework is built to improve grasping manipulation of the robot finger with RES, which would provide a self-adaptive control method to determine whether the objects are grasped successfully or not. Robot fingers integrated with soft sensors would promote the development of sensing and grasping abilities of the robot finger and interaction with human beings.

## 1. Introduction

Electronic skin (E-skin) with human-like sensory capabilities is widely applied to artificial intelligence [1], biomedical engineering [2,3], and the prosthetic hand [4]. E-skin with a tactile sensing function is a critical element for the robot to improve sensing ability [5,6], manipulation accuracy [7,8], and human–robot interaction [9,10]. To mimic the comprehensive tactile sensing capability and mechanical properties of human skin [11], flexible tactile sensors assembled to robots would improve the sensing and manipulation abilities with outside environment, robots, and human beings.

Robotic skin with conventional rigid sensors were reported to detect the bending status of the robot finger and the contact pressure when grasping objects [12]. Soft materials with high stretchability have been used for E-skin for human–machine interaction [13,14,15]. The elasticity and plasticity of viscoelastic hydrogels are highly frequency-dependent or strain-rate-dependent, which might deviate the performance of the sensors [16]. The strain-rate dependence of the hydrogel is proved by using cellulose nanofibrils (CNFs) as fillers in the polyacrylamide (PAAm) matrix for optimizing the electrical performance of the sensor with variance strain-rate [17]. An ionic hydrogel-based pressure sensor with high stretchability has been designed and fabricated for human motion detection [18,19]. Capacitive sensing pressure sensors with electric double layers (EDLs) based on ionic gels are designed for hand gesture detection [20]. Moreover, capacitive sensors for the tactile sensing of E-skin have demonstrated high strain sensitivity and compatibility with static force measurement [21]. Recently, robotic skins for biomimetic sensing were extended to tactile tensors for stable grasping applications [22,23]. A tactile sensor array with a thin strip of silicon rubber was applied to determine the contact location [24,25]. Robotic skin based on conducting rubber was utilized to detect slippage on its surface for grasping the objects with a significant decrease in the resistance of the rubber [26]. Both the variable resistance and capacitance for tactile sensors were important to the robotic skins integrated into the robot finger for grasping manipulation [27]. A negative force feedback control framework should be built for grasping manipulation of the robot finger with robotic electronic skin (RES) for improving the grasping capability.

RES is assembled onto the robot finger, which provides the similar tactile sensing capability of human skin for finger gesture detection. The strain and pressure sensor are integrated into the RES for recording the bending angle and contact pressure data. The strain sensor is integrated onto the back surface of the robot finger for recording the bending angle data of the robot finger. The pressure sensor is placed onto the finger tip for recording the pressure data when touching the objects. The main contribution of this work is as follows:(1)RES is designed with a simultaneous strain and pressure tactile sensing ability based on the soft conductive hydrogel. The soft conductive hydrogel is synthetized with several polymers through a sol–gel process for guaranteeing the overall softness of RES. RES is designed for finger gesture detection and grasping manipulation. RES with a strain and pressure sensor on the substrate is compatible with the robot finger, which would expand the application abilities of the robot finger in human–machine interaction (HMI) areas.(2)A negative force feedback is applied to RES for improving the grasping capability of the robot finger. RES is adhered onto the robot finger for bending strain and touching pressure recording during the manipulation of the robot finger. The negative force feedback control method is built for grasping manipulation of the robot finger integrated with RES, which would provide a self-adaptive control method to determine whether the objects are grasped successfully or not.

The rest of this paper is arranged as follows. The materials and methods of RES are illustrated in Section 2. Results and discussions for robot finger grasping manipulation based on RES is introduced in Section 3. Finally, the whole text is concluded.

## 2. Materials and Methods

### 2.1. Design of RES

Pressure and strain data are very important for the robot finger to interact with the outside environments, such as grasping objects, touching the obstacles, and recognizing the morphology of complex objects [28,29]. The strain sensor is assembled onto the back surface of the robot finger to collect the bending angle data of the robot finger. The pressure sensor is adhered onto the tip of the robot finger to collect the pressure data. The bending status of the robot finger is captured by the strain sensor, but it is difficult for the robot fingers to know whether the objects are grasped successfully or not with strain data [30]. The pressure sensor integrated into the tip of the robot finger is used as the negative feedback information to the robot fingers to determine whether the objects are grasped successfully. 

Figure 1a depicts the back and front views of the robot fingers integrated with the strain and pressure sensors. It is seen as a soft electronic skin, which is adhered onto the surface of the robotic finger (Figure 1b). The working principles of the two types of sensors are shown in Figure 1c. The resistance of the strain sensor varies with the bending of the robot finger, and the capacitance of the pressure sensor varies with the pressure applied at the fingertip. For satisfying the large deformation of the robot fingers, the strain and pressure sensor should also be deformed with the fingers. A soft conductive hydrogel with piezoresistive property is adopted to the strain sensor for the bending of the robot finger. Soft conductive hydrogel electrodes are also adopted to the capacitive pressure sensor, which is integrated onto the tip of the robot finger. RES with a strain and pressure function integrated into the robot finger would expand the sensing ability with the outside environment and promote the grasping ability.

### 2.2. Material Selection and Fabrication Process

The softness of the RES is critical for following the motion of the robot finger. The flexible polymer would be synthetized and adopted to the fabrication of RES [31]. Figure 2 depicts the main fabrication process of RES integrated into the robot finger, which includes preparation of the soft conductive hydrogel and sol–gel solution process for the conductive ionic hydrogel [32]. First, 15.62 g of acrylamide (AAm) (Aladdin Co., Shanghai, China) power and 16.029 g of NaCl are added into 100 mL of DI water. Next, 0.17 g of ammonium persulfate (AP) monomer (Aladdin Co., Shanghai, China) is added into the prepared solution, followed by the addition of 0.06g of chemical cross-linker *N*,*N*-methylenebisacrylamide (MBAA) (Aladdin Co., Shanghai, China) to enhance the conductivity of the conductive ionic gel. Afterward, 0.25g of *N*,*N*,*N*,*N*-tetramethylethylenediamine (TEMED) (Aladdin Co., Shanghai, China) are cross-linked for the ionic gel through the sol–gel process [33,34], which involves in situ polymerization and formation of a physically cross-linked network, as shown in Figure 2a. Carbon grease (847, MG Chemical Co., Burlington, ON, Canada) electrodes, as the conductive conductors, are printed onto the sensors for extracting the strain and pressure data of RES. Figure 2b depicts the schematic graph of the stretching and relaxing process of the fabricated transparent ionic gel. The ionic hydrogels still have good conductive ability during the stretching and relaxing process as the cross-link in ionic solution would suffer larger deformation.

RES integrated with a strain and pressure function is adhered onto the robot finger via double-sided tape for robot finger gesture monitoring. A soft conductive ionic hydrogel is adopted to RES for collecting strain and pressure data, which is used to mimic the tactile function ability. The strain sensor with ionic conductive gels is used to monitor the deformation of the robot finger. The capacitive pressure sensor with a dielectric layer (VHB 4910, 3M Co., Saint Paul, MN, USA.) and electrodes (conductive ionic hydrogel) have been designed to collect the tactile pressure data. Figure 3 depicts RES with strain and pressure sensors laminated to the surface of the robot finger. The optical image of RES with strain and pressure sensors on the polydimethylsiloxane (PDMS) substrate is shown in Figure 3a. The bending status of RES is shown in Figure 3b, which is bent to the finger-compatible shape for integration of the robot finger. Figure 3c shows the optical image of the robot finger integrated with RES to record the bending data and the pressure information of robot finger. Figure 3d,e show RES wears well with the robot finger, which can follow the bending actions of the robot finger.

### 2.3. Electrical Performance

Electrical stability is a vital property for long-term application of RES with strain and pressure sensors, as shown in Figure 4. For the strain sensor, the data acquisition system and signal analyzer (semiconductor character system, Keithley 4200-SCS, Keithley Co., Beaverton, OR, USA) are applied to measure the electrical performance combined with the home-made stretching jig. Figure 4a shows that the variable resistance has the linear relationship with the external applied strain. It is represented as: Δ*R*/*R_0_* = 9.98 ∗ *ε* (*R*^2^ = 0.99), where ε is the external applied strain to the sensor, *R*_0_ is the initial value of the strain sensor without deformation, and Δ*R* is the variable resistance value. The gauge factor (*GF*) is the paramount parameter for the strain sensor, which indicates the electrical response upon strain and is defined as $GF=\frac{\Delta R/R_{0}}{\Delta \varepsilon }$. The *GF* of the strain sensor (Figure 4a) is computed as 9.98, which is large and sensitive enough for hand gesture detection. Figure 4b depicts the long term monitoring of variable resistance value output under repeated loading and unloading process. It is validated that the stable resistance output (*GF*~9.98) is continuously read without obvious deviation under 200 cycles of the loading and unloading process. The stable properties of the soft strain sensor with conductive ionic gel would provide an effective approach to long-term hand gesture detection.

For the capacitive pressure sensor with hydrogel electrodes, the electrical performance is evaluated by the data acquisition system and signal analyzer (semiconductor character system, Keithley 4200-SCS, Keithley Co., Beaverton, OR, USA) and the loading cell. With the increasing loading value to the capacitive pressure sensor, the output capacitance value also increases, as shown in Figure 4c. The vertical blue dashed line represents the pressure data is 4.12 kPa. The capacitance value is $C=\frac{\varepsilon S}{d}$, where *S* is the opposite area of the capacitor plate, *d* is the distance between the capacitor electrodes, ε is the dielectric constant of the dielectric layer of the capacitor. It is found that the output capacitance has a non-linear response to the distance. Under external pressure to the capacitor, the distance between the capacitor electrodes would be reduced, and the output capacitance value would be increased. For analyzing the electrical performance of the capacitor easily, the approximate linear relationship with the loading pressure value $C=f(P)P+C_{0}$ is adopted to depict the electrical performance of the capacitor at one span of the load pressure. Sensitivity for the pressure sensor indicates the electrical response upon pressure as $\mathrm{Sensitivity}=\frac{\Delta C/C_{0}}{\Delta P}$, where Δ*C* is the variable capacitance value, *C*_0_ is the initial capacitance value, and P is the pressure loading. The piecewise linear function is adopted to fit the relationship between the capacitance value C and the pressure P. Below 4.12 kPa, the relationship between the capacitance value C and the pressure P is represented as *C* = 0.105 ∗ *P* + 5.72 (R^2^ = 0.95). At the span from 4.12 kPa to 15 kPa, *C* = 0.0327 ∗ *P* + 6.11 (R^2^ = 0.97). The sensitivity of the hydrogel-based capacitive pressure sensor is 0.105 kPa^−1^ below 3.61 kPa, and 0.0327 kPa^−1^ in the range from 4.12 to 15 kPa.

Figure 4d depicts long-term monitoring of the variable capacitance value output under the repeated loading and unloading process. It is validated that the stable capacitance output can be continuously read without obvious deviation, demonstrating the long-term stability of the sensor. It is revealed that the excellent reproducibility of resistance variations of conductive ionic gel appeared during loading and unloading cycles, strongly confirming the remarkable electrical stability. The electrical performance of RES has been evaluated by experiments for helping the manipulation behavior of the robot finger. The strain and pressure sensing functionalities are integrated into RES to provide more accurate gesture data to improve the motion ability of the robot finger.

## 3. Result and Discussions

### 3.1. Finger Gesture Detection

RES is assembled onto the robot finger for continuous pressure and strain data recordings, as shown in Figure 5. Figure 5a depicts the corresponding strain data under the different gestures of the robot fingers. It is shown that the bending status data of the robot finger is captured by the flexible strain sensor based on soft conductive hydrogel. The bending status of the index and middle two fingers generates four different finger gestures. The strain data variation of the robot finger is collected by the soft strain sensor of RES, and different hand gestures generate different control commands, which provide a potential application for the robot finger to interact with other robots. Figure 5b depicts the strain data recorded by RES with a repeated bending and releasing process with different bending angles. The amplitude of the strain data is similar during repeated bending with the same bending angles. The strain data from RES with the rotating engineering 180° is larger than the strain data with 90°. It is shown that electrical performance stability from RES is validated over 100 times with different turning angles.

Pressure sensor data is a tactile sensing function for the robotic finger to interact with the outside environment. Figure 5c depicts the variable capacitance value of the capacitive pressure sensor during the touching object process with different loadings. It is shown that the typical signals are recorded during the pressing process. Little, middle, large, and heavy pressures are applied to the objects via robot finger, and the typical pressure data is recorded by RES. The peak values increase with the increasing pressure loadings (little, middle, large, and heavy). Figure 5d depicts the output capacitance performance when the robot finger touches the soft rubber and the rigid wood. The red line represents the rigid wood and the black line represents the soft rubber. The output capacitance value when touching the rigid wood is larger than the output data from touching soft rubber. RES integrated with the robot fingers would not damage soft objects, which would provide the potential applications in grasping, manipulation of the robot finger, and biomedical applications.

### 3.2. Grasping Experiments

RES integrated into the robot finger is used to record the strain and pressure data for grasping the objects. Figure 6a depicts the experiment platform for the robot finger to grasp the objects, including the robotic finger, RES, data acquisition board, NI process controller, MATLAB/Simulink software, microcontroller STM32, and the objects. RES is used to collect the strain and pressure data during the bending process of the robot fingers, and flexible conductive films are connected to the carbon grease electrode of RES for extracting the sensing data for motion control of the robot finger. The strain and pressure data from RES is transmitted to the NI industry control computer, and the control commands are generated to control the motion of the robot finger by the microcontroller STM32. The capacitive pressure sensors provide an effective way to determine how to grasp the objects successfully (if the pressure data of the robot fingers is larger than 0). The closed loop control model for the robot grasping process with pressure negative feedback is illustrated in Figure 6b. The microcontroller STM32 generates a pulse-width modulation (PWM) wave to control the rotating angle of the steering engine. The robot finger bends to a certain angle for grasping the object when the robot finger makes contact with the objects. The capacitive pressure sensor is used to record the contract pressure at the fingertip/object interface. If the capacitive pressure data from RES is 0, the robot finger does not make contact with the target object. The robot finger still bends to a larger angle to grasp the objects until the pressure data from the soft capacitive sensor is larger than 0, and the object is grasped by the robot finger successfully.

The objects could be grasped successfully by the robot fingers integrated with RES with the control signal information flow (Figure 6b). Figure 7a depicts the optical images of the robot finger to grasp a plastic lid in the front and side views. The corresponding strain and pressure data distributions are depicted in Figure 7b with the pressure data negative feedback. At the starting time, the robot finger begins to bend for grasping the plastic lid, and the corresponding strain data is recorded by the soft strain sensor of RES. The pressure data is 0 as the robot finger does not touch the plastic lid. At time 2.2 s, the middle robot finger bends to a certain angle, and the robot fingertip touches the plastic lid as the pressure data from index robot finger is larger than 0. The strain data from the robot finger increases with the increasing bending angles of the robot fingers. The plastic lid is successfully grasped at time 2.9 s as the pressure data from middle, index, and ring fingers is larger than 0. These three robot fingers would stop bending as the pressure data is used as the feedback to the robot finger. The little finger continues to bend to grasp the plastic lid until the pressure from the little finger is larger than 0 at time 3.2 s. Both the strain and pressure data stay invariable from time 3.2 s to 4 s.

If the force negative control method is not adopted into the grasping of the plastic lid, the robot finger would not know whether the objects are grasped or not. It may damage the objects with larger bending angles of the robot finger, and it may also not grasp the objects successfully with smaller bending angles. Figure 7c depicts the strain and pressure data distribution from RES during the grasping of the plastic lid process without using force negative control methods. The blue dashed line in Figure 7 shows that the robot finger touch the object stalely with force negative feedback. It is shown that the strain data distribution with force negative feedback are similar to pressure and strain data distribution from RES with negative feedback before the pressure data is larger than 0 during the bending process of the robot finger for grasping the plastic lid. After the plastic lid is successfully grasped by the robot finger, the robot finger would still bend for grasping the objects, which would damage the plastic lid. It is found that the pressure data of RES from the robot finger would still increase from time 3.2 s to 4.0 s as the robot finger would continue to bend, which would increase the contact pressure at the capacitive pressure sensor–object interface.

Considering the motion ability of the robot finger, strain data induced by bending angles of the robot finger is recorded by RES, and the output value of the capacitive sensor is captured at the object–RES interface under the grasping process. The feedback control method for the grasping manipulation of the robot finger is built with strain and pressure data from RES. It has been validated that smaller pressure is required for the robot finger to grasp objects with negative force feedback of RES, which promotes the grasping of objects by the robot finger with more accurate data and feedback control methods. The objects would be successfully grasped by the robot finger with RES using less energy consumption with negative force feedback control compared to the case without negative force feedback.

A robotic electronic skin based on other sensors are used for finger gesture detection, grasping manipulation, and HMI applications. Several effects (bendability, stretchability, mounting method, and control strategy) are adopted to evaluate the performances of robotic electronic skin for controlling the bending and grasping abilities of the robot finger, as shown in Table 1. The robot finger integrated with rigid strain/pressure sensors are designed for hand rehabilitation and surgery application. The rigid strain/pressure sensors are used to detect the finger gesture which would damage the objects [7]. A multisensory sensor system with pressure, temperature, and humidity function is deployed to the robotic finger for accomplishing complicated tasks. However, the conventional rigid sensors could not follow the deformation of the hand bending actions. It would be not compatible with the soft skin, especially in the case of bended fingers [25]. The dual-mode sensor array is also implemented as E-skin in a bionic hand for tactile sensing with high pressure range (10–120 kPa) and high sensitivity (highest 1.04 V/kPa) at low pressure range (<10 kPa). However, robotic electronic skin could not follow the motion of robotic skin [3]. Robot-finger-compatible RES with strain and pressure sensor units is developed for finger gesture detection and grasping manipulation. It could follow the motion of the robot finger for more accurate strain and pressure data with high sensitivity: gauge factor ~9.98, pressure sensor ~0.105 kPa^−1^ below 3.61 kPa, and 0.0327 kPa^−1^ in the range from 4.12 to 15 kPa, which is enough sensitivity to detect the finger bending actions and grasping of objects. The negative force feedback control framework is built for grasping manipulation of the robot finger. Experiments demonstrated that RES integrated into the robot finger would expand the motion accuracy and grasping of objects, which is helpful to the robot finger to complete more complicated tasks.

## 4. Conclusions

RES integrated with a strain and pressure sensor based on transparent ionic conductive gel has been designed for gesture detection and grasping manipulation of the robot finger. Therefore, the mechanical and electrical performances of RES would be promoted in several aspects for robot grasping manipulation. RES was fabricated and assembled by the functional materials through a sol–gel process for guaranteeing the overall softness of RES. The strain sensor was integrated onto the back surface of the robot finger to collect the bending angle data of the robot finger. The pressure sensor was integrated onto the fingertip to collect the pressure data. RES was assembled onto the robot finger using double-sided tape, which provided the similar tactile sensing capability of human skin for finger gesture detection and grasping manipulation. The negative force feedback control framework was built for grasping manipulation of the robot finger with RES, which would provide an adaptive control method to determine whether the objects were grasped successfully or not. RES with flexible tactile sensors was applied to promote the grasping and sensing ability of the robot finger with the outside environment, robots, and human beings, such as human–human interaction.

## Figures and Tables

**Figure 1 polymers-14-03930-f001:**
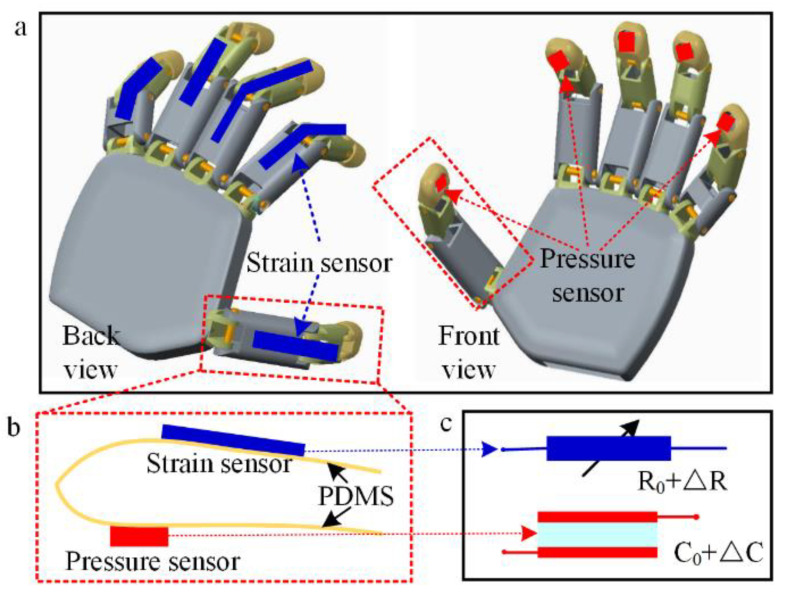
Schematic graph of RES. (**a**) The robot finger integrated with stain and pressure sensors. (**b**) Strain and pressure sensor on the soft substrate. (**c**) Principle of the strain and pressure sensors.

**Figure 2 polymers-14-03930-f002:**
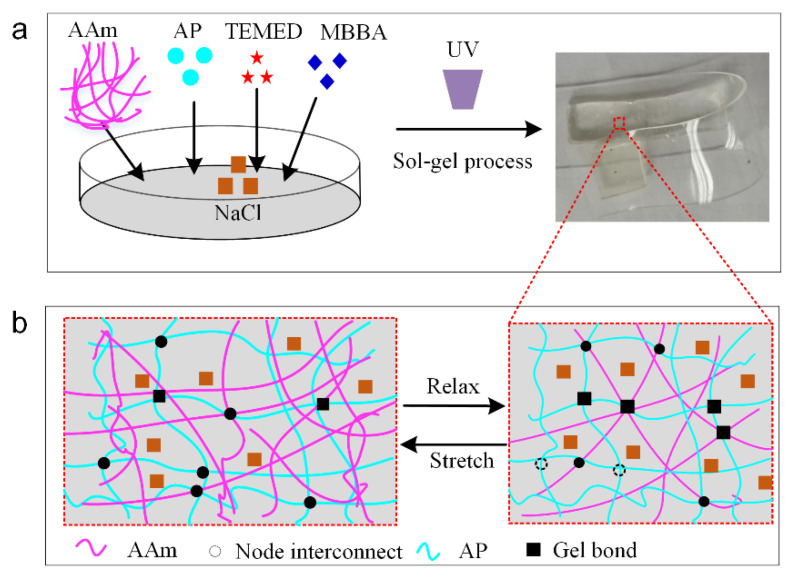
Fabrication process of RES. (**a**) Sol–gel process for conductive ionic gel fabrication. (**b**) Schematic graph of the stretching and relaxing process of the conductive ionic hydrogel.

**Figure 3 polymers-14-03930-f003:**
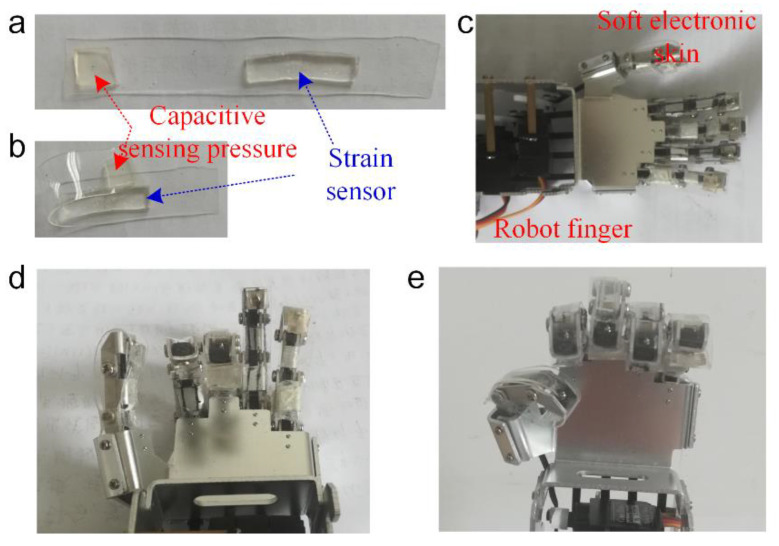
Integration of RES to the robot finger. Optical image of (**a**) RES on the PDMS substrate. (**b**) RES bending to the finger-compatible shape. (**c**) The robot finger with RES. (**d**,**e**) Optical images of the different bending actions of the robot finger with RES.

**Figure 4 polymers-14-03930-f004:**
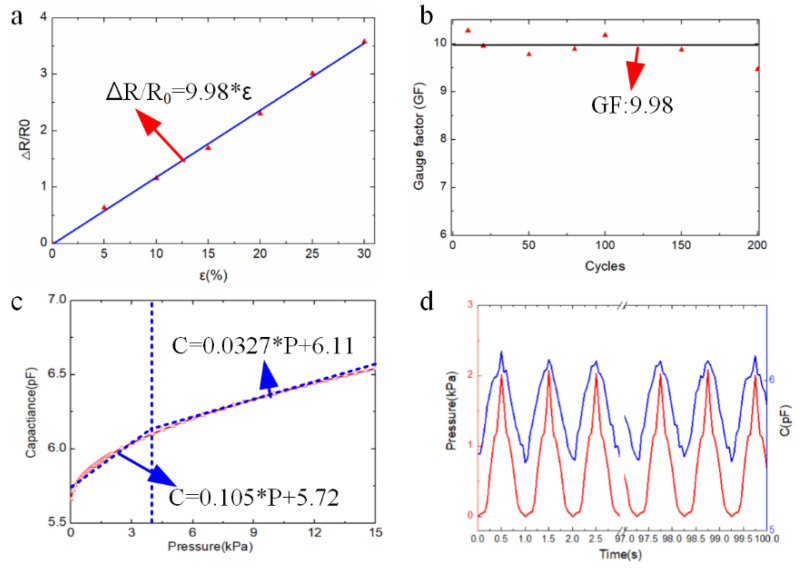
Electrical performance of RES. (**a**) Variable resistance value versus the strain. (**b**) Repeated strain data with different loading cycles. (**c**) Relationship between the capacitance and external pressure. (**d**) Repeated capacitance data with different loading cycles.

**Figure 5 polymers-14-03930-f005:**
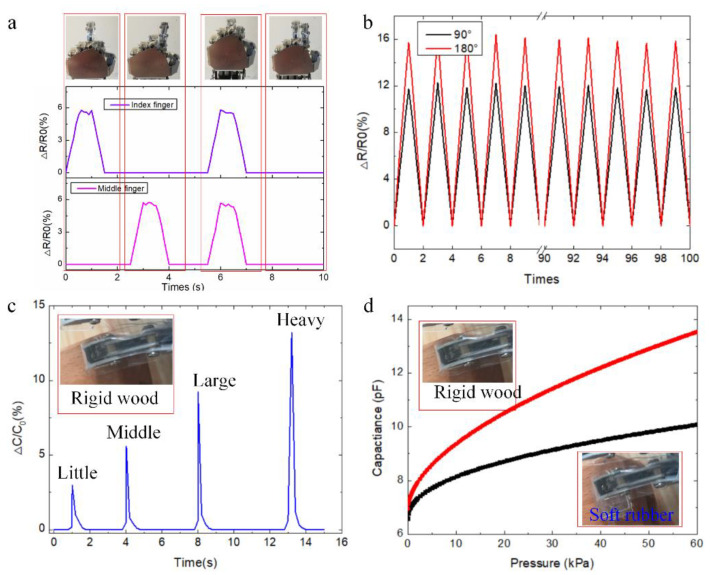
Demonstration experiments of RES for collecting motion data of robotic finger. (**a**) Typical strain data under different robot finger gestures. (**b**) Continuous strain data with repeated bending and releasing process. (**c**) Output capacitance of the pressure sensor with different loadings. (**d**) Pressure data is recorded when touching soft rubber and rigid wood.

**Figure 6 polymers-14-03930-f006:**
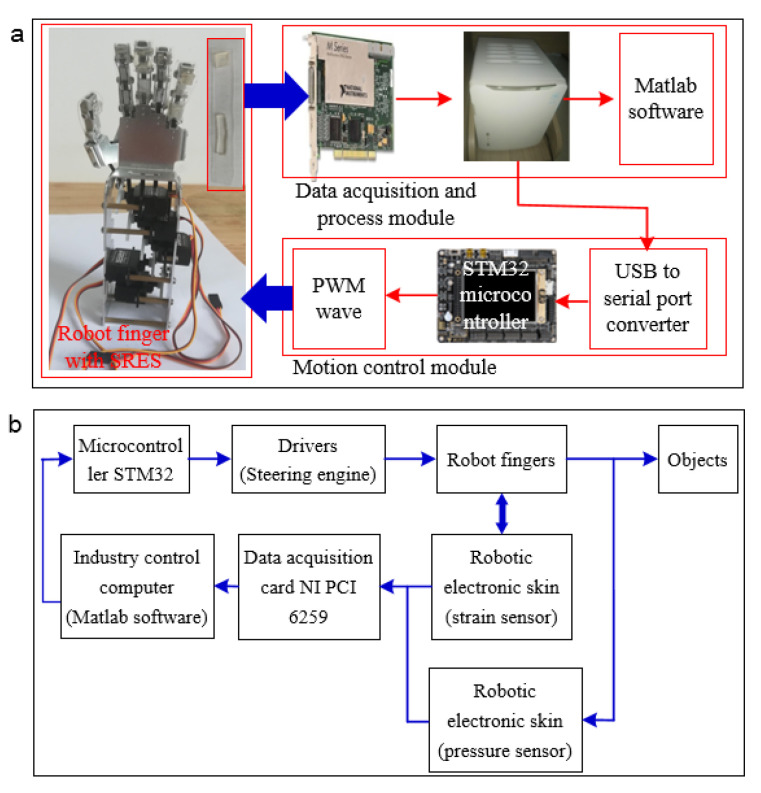
(**a**) Experiment platform of the robot finger with RES to grasp objects. (**b**) Signal flow of the grasping process of the robotic finger with negative force feedback by the capacitive pressure sensor.

**Figure 7 polymers-14-03930-f007:**
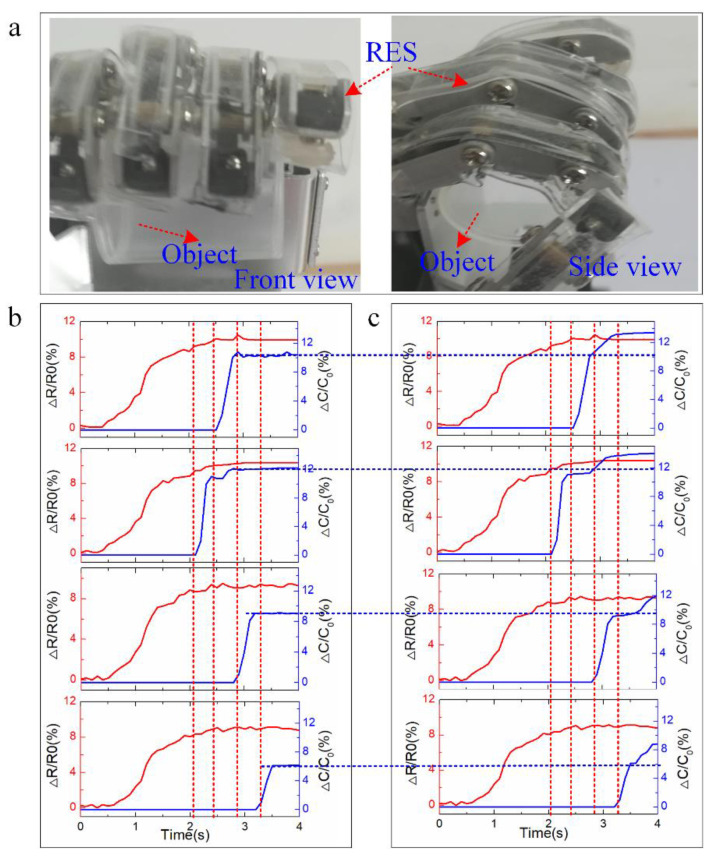
Strain data distribution during the grasping process of the robotic finger with force negative feedback. (**a**) Optical image of robot finger with RES for grasping plastic lid; Strain and pressure distribution during the grasping of the plastic lid process (**b**) with force negative feedback and (**c**) without force negative feedback.

**Table 1 polymers-14-03930-t001:** Performances of robotic electronic skin (RES).

Robot Skin	Bending Radius	Stretchability	Mounting Method	Feedback
Literature [7]	>10	~0.3%	Mechanical assembly	Yes
Literature [25]	>5 mm	<1%	Bandage	Yes
Literature [3]	<2 mm	>10%	Not illustrated	No
This paper	<1 mm	>15%	Tape	Yes

## Data Availability

Not applicable.

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
