# Peer review of "Soft Conductive Hydrogel-Based Electronic Skin for Robot Finger Grasping Manipulation"

_polymers, 2022, doi:10.3390/polym14193930_

Round 1

Reviewer 1 Report

1. The manuscript should be reviewed for typos, such as lines...62, 126, 214, 229, 232, 238, 265, 268, 272, 313...;

2. Since it is difficult for the reader to separate the "Introduction" section from the experimental part performed in this study, I recommend that chapters 2&3 be introduced in the "Materials and methods" section;

3. Lines: 109-111 The authors are requested to revise the sentence "16.62 g acrylamide...(MBAA)";

4. Line 121: What does "PAP" represent in figure 2b?

5. Lines 133&139: What is "finger-tupe"?

6. Line 146: for consistency, use R0(0-subscript) and R2(2-superscript)

7. Line 223-226 & 235-236 The authors are requested to revise the sentences

8. line 261 remove ”are” from ”distribution are...”

9. Line 297 remove ”is”

10. Line 307: Chapter 5 seems to be dedicated only to the conclusions. I recommend renaming the chapter 4 to "Results and discussion", and the conclusions chapter to be self-standing. I also, recommend, reviewing the conclusions highlighting the most important results obtained in this study.

11. Lines 308-309 The paragraph should be deleted. It seems to be from the "Guide for authors" form.

Author Response

Response to Reviewer 1 Comments

Point 1: The manuscript should be reviewed for typos, such as lines...62, 126, 214, 229, 232, 238, 265, 268, 272, 313...;

Response 1: Thanks to the reviewer for pointing out the spelling mistakes.

All the spelling mistakes have been revised highlighted in yellow color. And some other spelling mistakes are checked throughout the manuscript.

Point 2: Since it is difficult for the reader to separate the "Introduction" section from the experimental part performed in this study, I recommend that chapters 2&3 be introduced in the "Materials and methods" section;

Response 2: Thanks to the reviewer for proposing useful suggestions.

Chapters 2&3 are combined to "Materials and methods" section. The section "Materials and methods", the subsections “2.1. Design of RES”, “2.2. Material selection and fabrication process” and “2.3. Electrical performance” are highlighted in yellow color.

Point 3: Lines: 109-111 The authors are requested to revise the sentence "16.62 g acrylamide...(MBAA)";

Response 3: Thanks to the reviewer for pointing out the mistakes.

“15.62g acrylamide (AAm) (Aladdin Co. China) power and 16.029g NaCl are added into 100 ml DI water, 0.17g ammonium persulfate (AP) monomer (Aladdin Co. China) and the addition of 0.06g chemical crosslinker N,N-methylenebisacrylamide (MBAA) (Aladdin Co. China).” has been revised to “15.62g acrylamide (AAm) (Aladdin Co. China) power and 16.029g NaCl are added into 100 ml DI water, and 0.17g ammonium persulfate (AP) monomer (Aladdin Co. China) , the addition of 0.06g chemical crosslinker N,N-methylenebisacrylamide (MBAA) (Aladdin Co. China) are also added to the solution for the fabrication of the conductive ionic gel.”

And it is highlighted in page 3 in yellow color.

Point 4: Line 121: What does "PAP" represent in figure 2b?

Response 4: Thanks to the reviewer for proposing useful suggestions.

PAP has been revised to “ammonium persulfate (AP)”, and Figure 2b is updated in Page 3.

Point 5: Lines 133&139: What is "finger-tupe"?

Response 5: Thanks to the reviewer for pointing out the mistakes.

“Finger-tupe” has been changed into “Finger-compatible”, and it is highlighted in Page 4 lines 133, 138-139 in yellow color.

Point 6: Line 146: for consistency, use R0(0-subscript) and R2(2-superscript)

Response 6: Thanks to the reviewer for pointing out the mistakes.

R0(0-subscript) has been revised to “R0”, and R2 has been revised to “R2” in Page 4 highlighted in yellow color.

Point 7: Line 223-226 & 235-236 The authors are requested to revise the sentences

Response 7: Thanks to the reviewer for pointing out the mistakes.

“RES is used to collect the strain and pressure data as the bending process of the robot fingers by the data acquisition board which is connected to the carbon grease of RES using the flexible conductive films for motion control of the robot finger. The strain and pressure data from RES would be processed by NI industry control computer which the commands would be sent to the microcontroller STM32 to control the motion of robot finger. “ has been changed to “RES is used to collect the strain and pressure data during the bending process of the robot fingers, and flexible conductive films are connected to the carbon grease electrode of RES for extracting the sensing data for motion control of the robot finger. The strain and pressure data from RES would be transmitted to NI industry control computer, and the control commands are generated to control the motion of robot finger by the microcontroller STM32.” And it is highlighted in Page 6 in yellow color.

“If the pressure data from fingertip captured by the robot finger is 0, the robot finger would still bend to a larger angle to grasp the objects until the pressure data from the soft capacitive sensor is larger than 0, which the object is grasped by the robot finger successfully.” has been changed to “If the capacitive pressure data from RES is 0, robot finger would not contact with the target object. The robot finger would still bend to a larger angle to grasp the objects until the pressure data from the soft capacitive sensor is larger than 0, and the object is grasped by the robot finger successfully.” And it is highlighted in Page 7 in yellow color.

Point 8: line 261 remove ”are” from ”distribution are...”

Response 8: Thanks to the reviewer for pointing out the mistakes.

“It is shown that the strain data distribution with force negative feedback are similar to pressure and strain data distribution from RES with negative feedback” has revised to “It is shown that the strain data distribution with force negative feedback are similar to pressure and strain data distribution from RES with negative feedback” in Page 7 in line 261 highlighted in yellow color.

Point 9: Line 297 remove ”is”

Response 9: Thanks to the reviewer for pointing out the mistakes.

“This manuscript illustrates that a robot-finger-compatible RES with strain and pressure function is designed to finger gesture detection and grasping manipulation.” has changed to “The manuscript illustrates that a robot-finger-compatible RES with strain and pressure function for finger gesture detection and grasping manipulation.” in Page 9 Line 296 highlighted in yellow color

Point 10: Line 307: Chapter 5 seems to be dedicated only to the conclusions. I recommend renaming the chapter 4 to "Results and discussion", and the conclusions chapter to be self-standing. I also, recommend, reviewing the conclusions highlighting the most important results obtained in this study.

Response 10: Thanks to the reviewer for proposing useful suggestions.

Chapter 4 (Now Chapter 3) is revised to section "Results and discussion".

The conclusions chapter are also changed in Page 9 highlighted in yellow color.

Point 11: Lines 308-309 The paragraph should be deleted. It seems to be from the "Guide for authors" form.

Response 11: Thanks to the reviewer for pointing out the mistakes.

The paragraph “Authors should discuss the results and how they can be interpreted from the perspective of previous studies and of the working hypotheses. The findings and their implications should be discussed in the broadest context possible. Future research directions may also be highlighted.” has been deleted from the manuscript.

Reviewer 2 Report

The manuscript presented a decent work on strain and stress sensors applied to robotic fingers. With the combination of two sensors, anti-smashing function was realized. The work finished thoroughly which can provide great reference to the robotic field. The reviewer has some comments. Once the reviewer would like to recommend the manuscript for publication, once the comments explained. The detail lists below:

1.       The sensors’ working mechanism needs to be stated in detail. The strain sensor was clear. The change of resistance is linearly proportional to the length. But, for the capacitor sensor, it seems like the response approached to a non-linear region where the capacitance should be inverse proportional to the thickness of the capacitor. The origin of the non-linearity needs to be discussed in detail. Is it caused by the non-linearity in the electrical response of the capacitor, or from the non-linearity of the stress-strain response on the materials?

2.       The reviewer suggests the authors to post some strain-resolution and stress-resolution of the two sensors, measurements or theoretical predictions, which is important for system designs in terms of scales and potential applications.

3.       Also, if possible, the discussion or estimation of the maximum strain or stress needs to be stated in the manuscript.

4.       Hydrogel is highly viscoelastic. The elasticity and plasticity of hydrogels are highly frequency dependent or strain rate dependent which might deviate the performance of the sensors. (https://doi.org/10.3390/polym12071462) and (https://doi.org/10.1021/acs.langmuir.9b01532). The strain rate dependence of the involved hydrogel ideally should be determined. If the authors had some data regarding to that, please include. Otherwise, please discuss more about the predicted sensor performance with variance strain rate.

5.       The size or resolution of the figures definitely needs to be improved, which are barely clear in the current version.

Author Response

Response to Reviewer 2 Comments

Point 1:  The sensors’ working mechanism needs to be stated in detail. The strain sensor was clear. The change of resistance is linearly proportional to the length. But, for the capacitor sensor, it seems like the response approached to a non-linear region where the capacitance should be inverse proportional to the thickness of the capacitor. The origin of the non-linearity needs to be discussed in detail. Is it caused by the non-linearity in the electrical response of the capacitor, or from the non-linearity of the stress-strain response on the materials?

Response 1: Thanks to the reviewer for proposing the useful suggestions.

The output capacitance value C=εS/d, S is the opposite area of the capacitor plate, d is the distance between the capacitor electrodes, ε is dielectric constant of dielectric layer of the capacitor. It is found that the output capacitance has a non-linear response to the distance. Under external pressure to the capacitor, the distance between the capacitor electrodes would be reduced, and the output capacitance value would be increased. For analyzing the electrical performance of the capacitor easily, approximate linear relationship with the loading pressure value:  is adopted to depict the electrical performance of the capacitor at one span of the load pressure. The non-linearity in the electrical response of the capacitor is caused from the distance between the capacitor electrodes, which is affected by the external load to the capacitor during the grasping manipulation process. It is revised in Page 5 highlighted in green color.

Point 2: The reviewer suggests the authors to post some strain-resolution and stress-resolution of the two sensors, measurements or theoretical predictions, which is important for system designs in terms of scales and potential applications.

Response 2: Thanks to the reviewer for proposing useful suggestions.

For the strain sensor, data acquisition system and signal analyzer (Semiconductor character system, Keithley 4200-SCS, Keithley Co., US) is applied to measure the electrical performance combined with the home-made stretching jig. The variable resistance has the linear relationship with the external applied strain. GF of the strain sensor (Fig.4a) is computed as 9.98, which is large enough to hand gesture detection. Fig.4b depicts that the long term monitoring of variable resistance value output under repeated loading and unloading process. It is validated that the stable resistance output (GF ~ 9.98) is continuously read without obvious deviation under 200 cycles of the loading and unloading process. It is revised in Page 4 highlighted in green color.

For the capacitive pressure sensor with hydrogel electrodes, the electrical performance is evaluated by data acquisition system and signal analyzer (Semiconductor character system, Keithley 4200-SCS, Keithley Co., US) and the loading cell. The capacitance value  has a non-linear response to the distance. Under external pressure to the capacitor, the distance between the capacitor electrodes would be reduced, and the output capacitance value would be increased. For analyzing the electrical performance of the capacitor easily, approximate linear relationship with the loading pressure value:  is adopted to depict the electrical performance of the capacitor at one span of the load pressure. Piecewise linear is adopted to fit the relationship between the capacitance value C and the pressure P. Below 4.12 kPa, the relationship between the capacitance value C and the pressure P is represented as: C=0.105*P+5.72 (R2=0.95). At the span from 4.12 kPa to 15 kPa, C=0.0327*P+6.11 (R2=0.97). The sensitivity of hydrogel-based pressure sensor is 0.105 kPa-1 below 3.61 kPa, and 0.0327 kPa-1 in the range from 4.12 to 15 kPa. It is revised in Page 4 highlighted in green color.

Point 3: Also, if possible, the discussion or estimation of the maximum strain or stress needs to be stated in the manuscript.

Response 3: Thanks to the reviewer for proposing useful suggestions.

The maximum strain of the conductive ionic hydrogel could reach 100% reported in the previous work. However, the strain sensor is designed and fabricated to detect the bending process of the robot finger, and 30% (Figure 4a) stretchability of the strain sensor based on conductive ionic gel is measured and discussed in the manuscript. It is enough to detect the bending angle of robot finger with 30% stretchability.

Point 4: Hydrogel is highly viscoelastic. The elasticity and plasticity of hydrogels are highly frequency dependent or strain rate dependent which might deviate the performance of the sensors. (https://doi.org/10.3390/polym12071462) and (https://doi.org/10.1021/acs.langmuir.9b01532). The strain rate dependence of the involved hydrogel ideally should be determined. If the authors had some data regarding to that, please include. Otherwise, please discuss more about the predicted sensor performance with variance strain rate.

 Response 4: Thanks to the reviewer for proposing useful suggestions.

The elasticity and plasticity of viscoelastic hydrogels are highly frequency dependent or strain rate dependent which might deviate the performance of the sensors [16]. The strain rate dependence of the hydrogel is proved by using cellulose nanofibrils (CNFs) as fillers in the polyacrylamide (PAAm) matrix for optimizing the electrical perfor-mance of the sensor with variance strain rate [17].

References

  1. Y. Jin, T. Yang, S. Ju, H. Zhang, T.-Y. Choi, and A. Neogi, "Thermally Tunable Dynamic and Static Elastic Properties of Hy-drogel Due to Volumetric Phase Transition," Polymers, vol. 12, p. 1462, 2020.
  2. J. Yang, C. Shao, and L. Meng, "Strain Rate-Dependent Viscoelasticity and Fracture Mechanics of Cellulose Nanofibril Com-posite Hydrogels," Langmuir, vol. 35, pp. 10542-10550, 2019/08/13 2019.

Point 5: The size or resolution of the figures definitely needs to be improved, which are barely clear in the current version.

Response 5: Thanks to the reviewer for pointing out the deficiency.

Figures 3, 5 and 7 are updated with high resolution in manuscript.

Reviewer 3 Report

This manuscript is interesting and suitable for this Journal.
It is a manuscript with an interesting theme and of great translation. The authors carry out in an appropriate way and with recent references, an introduction to the state of the art that justifies the study and allows to see the novelty of the study.
The authors make a correct methodological description, with references that allow a correct reproducibility of the study and allows to see the veracity of the study.
The results are displayed correctly, although the authors must improve minor points such as the description of the figure legends. The reader can correctly follow the study with a correct causal discussion that justifies the results.
The conclusions are fair and are supported by these very interesting results.
I would suggest the authors improve the description of the figures and make a graphic summary.

The manuscript is adequate, but the manuscript must improve the use of English grammar.

Author Response

Response to Reviewer 3 Comments

Point 1:  The authors must improve minor points such as the description of the figure legends. The reader can correctly follow the study with a correct causal discussion that justifies the results.

Response 1: Thanks to the reviewer for proposing useful suggestions.

The description of the figure legends is updated carefully. The discussion is also revised to justify the results.

It has been revised in Pages 3, 4, 6, and 8 highlighted in brown color.

Point 2:  I would suggest the authors improve the description of the figures and make a graphic summary.

Response 2: Thanks to the reviewer for proposing useful suggestions.

The description of the figures is revised for justifying the results. It has been revised in Pages 3, 4, 6, and 8 highlighted in brown color.

Point 3:  The manuscript is adequate, but the manuscript must improve the use of English grammar.

Response 3: Thanks to the reviewer for proposing useful suggestions.

English grammar of the manuscript has been checked throughout the manuscript. It has been revised highlighted in yellow or brown color.

Round 2

Reviewer 2 Report

The authors addressed the comments from the reviewer very well. The reviewer suggests the acceptance of the manuscript. 

Author Response

Response to Reviewer 2 Comments

Point 1:  The authors addressed the comments from the reviewer very well. The reviewer suggests the acceptance of the manuscript. 

Response 1: Thanks to the reviewer for proposing the encouraging suggestions.

We will check the manuscript carefully for the publication to the journal.